



# LUCAS Cover photos 2006-2018 over the EU: 874,646 spatially distributed geo-tagged close-up photos with land cover and plant species label

Raphaël d'Andrimont[1], Momchil Yordanov[1], Laura Martinez-Sanchez[1], Peter Haub[2], Oliver Buck[3], Carsten Haub[3], Beatrice Eiselt[4], and Marijn van der Velde[1]

[1]European Commission Joint Research Centre (JRC), Ispra, Italy
[2]Imaging Consulting, Altlussheim, Germany
[3]EFTAS, EFTAS Fernerkundung Technologietransfer GmbH, Münster, Germany
[4]European Commission, Eurostat (ESTAT), Luxembourg, Luxembourg

**Correspondence:** Raphaël d'Andrimont (raphael.dandrimont@ec.europa.eu) and Marijn van der Velde (marijn.van-der-velde@ec.europa.eu)

**Abstract.** In the European Union, a tri-annual surveyed sample collects land cover and land use information under the Land Use/Cover Area frame Survey since 2006. A total of 1,351,293 observations at 651,780 unique locations for 106 variables along with 5.4 million landscape and point photos were collected during five LUCAS surveys. In addition to these photos, a set of previously unpublished LUCAS Cover photos were also taken, i.e. following the protocol, a close-up view of the tree, crop, and plant species. These photos contain more details so that tree, crop, and plant species should be identifiable. Between 2006 and 2018, 875,661 LUCAS Cover photos that show the relevant land cover in its entirety were collected. Due to surveyor differences, the images sometimes display elements that require a two-stage deep learning anonymisation process, after which 346 photos were removed before publication. This paper summarizes the collection of LUCAS Cover photos, the filtering for mandatory privacy issues, and provides links to download the data along with the photo metadata, and cross-links to the corresponding LUCAS harmonised survey data. Moreover, after presenting the final public and open dataset consisting in 874,646 photos , potential applications relying on recent advances in geo-spatial analysis and statistical learning such as large scale biodiversity monitoring are discussed.

## 1 Introduction

In the European Union (EU), a tri-annual surveyed sample of land cover and land use has been collected since 2006 under the Land Use/Cover Area frame Survey (LUCAS) (Gallego and Delincé, 2010). LUCAS has been carried out in 2006, 2009, 2012, 2015, 2018 and is planned for 2022. During the five campaigns already carried out, a total of 1,351,293 points at 651,780 unique locations were surveyed along with 5.4 million landscape photos. On each of these surveyed points, depending on the year observations were recorded on up to 109 variables. The combination of the information collected in the five LUCAS surveys has resulted in the most comprehensive in-situ database on land cover and land use in the EU (d'Andrimont et al. (2020)).





In addition to the landscape and point photos already published (d'Andrimont et al., 2020), other specific photos were taken including the LUCAS Cover photos, a close-up view of the land cover on which plant species should be identifiable. This photo was not taken to be published but to support visual quality control simultaneously along the field survey. Between 2006 and 2018, 875,661 of such LUCAS Cover photos were collected. However, as this specific LUCAS Cover photo was not designed
as an output of the survey, it has not been published yet.

The objective of this paper is to make this rich dataset available in analysis ready form to the research community for various use cases. The pre-requisite for using the LUCAS Cover data and photos in other applications (e.g. biodiversity monitoring, or machine readable calibration sources for EO) requires organizing, curating, documenting and publishing the photos following FAIR (Findability, Accessibility, Interoperability, and Reuse) principles (Wilkinson et al., 2016). This paper summarizes the
collection of LUCAS Cover photos, the filtering for mandatory privacy issues, and provides links to download the data along with the photo metadata, and cross-links to the corresponding LUCAS harmonised survey data.

## 2  In-situ LUCAS Survey protocol

LUCAS is a two phase sample survey. The first sample is a systematic selection of points on a grid with a 2 km spacing in Eastings and Northings covering the whole EU territory (Gallego and Bamps, 2008). Currently, it includes around 1.1 million
points (Figure 1) and is referred to as the master sample. Each point of the first phase sample is classified into one of ten land-cover classes via visual interpretation of ortho-photos or satellite images (ESTAT, 2018). Then a stratified sample is selected to obtain the desired statistically representative spatial distribution of sampled land cover classes according to the first phase visual interpretation (Scarnò et al., 2018).

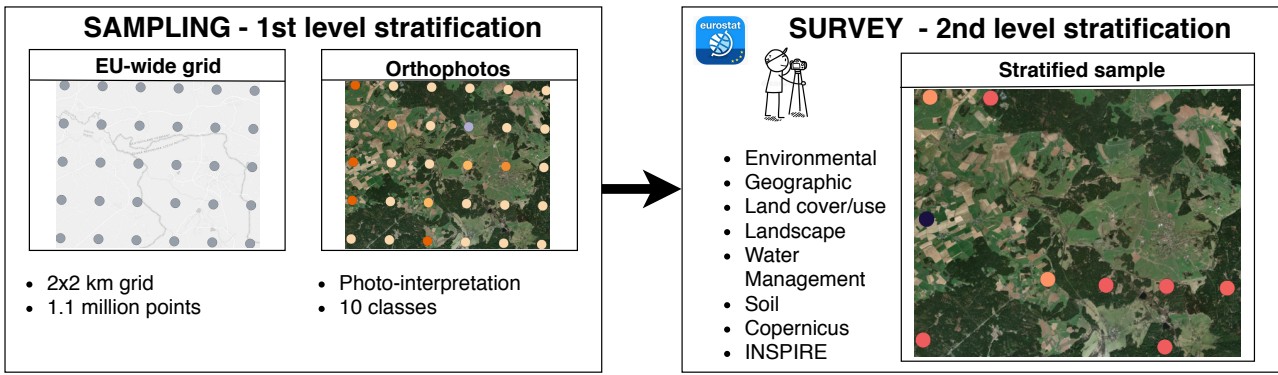

**Figure 1.** Schematic overview of the LUCAS and harmonisation methodologies. This illustrates the sampling at the basis of the production of the LUCAS primary data. LUCAS Cover photos are close-up photos originally collected to support the surveyors' interpretation and control.

## 3 LUCAS COVER photo collection protocol

As described in Eurostat (2018), LUCAS surveyors document their observations also in several sets of photos. The type of photo to be taken depends on the type of observation, the land cover, the presence or absence of water management, the need to collect a soil sample, and the need to document conflicting cases. Cardinal direction photos are taken for each observed in-situ point into Point, North, East, South, and West (P, N, E, S, W) direction (example of these are the five first photos from left to right in Figure 2). This P, N, E, S, W photo dataset corresponds to 5,440,459 photos for the five surveys. These photos are

publicly available for download along with an EXIF database in d'Andrimont et al. (2020), containing image metadata.

However, as described in Eurostat (2018), other non-publicly available photos were taken. Among these photos, the LUCAS Cover (C) photos were collected mainly from Cropland (class B), Woodland (class C), Shrubland (class D), and Grassland (class E). The aim of these cover photos is to enable the identification of the recorded crops and plants during simultaneous quality controls in the office by means of the photo on screen (Eurostat, 2018). The cover photo should be taken at a close

distance, so that the structure of e.g. leaves, barks, flowers or fruits can be clearly seen. See an example in Figure 2 on the right.

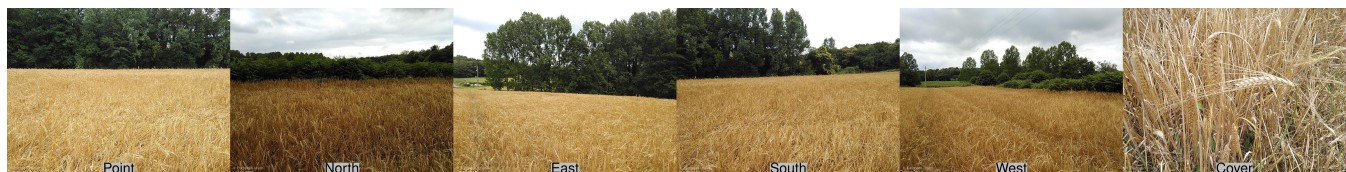

**Figure 2.** Example of LUCAS photos collected on a LUCAS point located in France in a barley field (lat:48.1645, long: -2.4970). For each LUCAS point, photos are collected for North, East, South, West, Point, and Cover.

## 4 Photo metadata extraction

The LUCAS Cover photos were obtained from the Eurostat archive via portable hard drives. The photos' metadata were then extracted with the ExifTool (v 12.10) (Harvey (2013)) resulting in a database of photos that was compared for completeness with the survey data records. The EXIF metadata were extracted for 82 fields (Table 1). Finally, the LUCAS cover EXIF table

was joined to the LUCAS harmonised database to provide all survey information into one unique table.



**Table 1.** The LUCAS Cover data set is provided with two tables, the EXIF table with 82 metadata attributes extracted from the photos along with the LUCAS harmonised table containing 121 attributes.

| Origin of attributes | # | Attribute names |
|---|---|---|
| Exif fields | 82 | ApertureValue, BrightnessValue, ColorSpace, ComponentsConfiguration, CompressedBitsPerPixel, Contrast, Copyright, CustomRendered, DateTime, DateTimeDigitized, DateTimeOriginal, DeviceSettingDescription, DigitalZoomRatio, ExifImageLength, ExifImageWidth, ExifOffset, ExifVersion, ExposureBiasValue, ExposureIndex, ExposureMode, ExposureProgram, ExposureTime, FileSource, Flash, FlashPixVersion, FNumber, FocalLength, FocalLengthIn35mmFilm, FocalPlaneResolutionUnit, FocalPlaneXResolution, FocalPlaneYResolution, GainControl, Gamma, GPSAltitude, GPSAltitudeRef, GPSDate, GPSInfo, GPSLatitude, GPSLatitudeRef, GPSLongitude, GPSLongitudeRef, GPSMapDatum, GPSSatellites, GPSTimeStamp, GPSVersionID, ImageDescription, InteroperabilityIndex, InteroperabilityOffset, InteroperabilityVersion, ISOSpeedRatings, LightSource, Make, MaxApertureValue, MeteringMode, Model, OECF, OffsetSchema, Orientation, Padding, PrimaryChromaticities, Rating, RelatedImageLength, RelatedImageWidth, ResolutionUnit, Saturation, SceneCaptureType, SceneType, SensingMethod, Sharpness, ShutterSpeedValue, Software, SubjectDistanceRange, WhiteBalance, WhitePoint, XResolution, YCbCrCoefficients, YCbCrPositioning, YResolution, year, pointid, file_path_ftp_cover, id |
| LUCAS HARMO fields | 121 | id, point_id, year, nuts0, nuts1, nuts2, nuts3, th_lat, th_long, office_pi, ex_ante, survey_date, car_latitude, car_ew, car_longitude, gps_proj, gps_prec, gps_altitude, gps_lat, gps_ew, gps_long, obs_dist, obs_direct, obs_type, obs_radius, letter_group, lc1, lc1_label, lc1_spec, lc1_spec_label, lc1_perc, lc2, lc2_label, lc2_spec, lc2_spec_label, lc2_perc, lu1, lu1_label, lu1_type, lu1_type_label, lu1_perc, lu2, lu2_label, lu2_type, lu2_type_label, lu2_perc, parcel_area_ha, tree_height_maturity, tree_height_survey, feature_width, lm_stone_walls, crop_residues, lm_grass_margins, grazing, special_status, lc_lu_special_remark, cprn_cando, cprn_lc, cprn_lc_label, cprn_lc1n, cprnc_lc1e, cprnc_lc1s, cprnc_lc1w, cprn_lc1n_brdth, cprn_lc1e_brdth, cprn_lc1s_brdth, cprn_lc1w_brdth, cprn_lc1n_next, cprn_lc1s_next, cprn_lc1e_next, cprn_lc1w_next, cprn_urban, cprn_impervious_perc, inspire_plcc1, inspire_plcc2, inspire_plcc3, inspire_plcc4, inspire_plcc5, inspire_plcc6, inspire_plcc7, inspire_plcc8, eunis_complex, grassland_sample, grass_cando, wm, wm_source, wm_type, wm_delivery, erosion_cando, soil_stones_perc, bio_sample, soil_bio_taken, bulk0_10_sample, soil_blk_0_10_taken, bulk10_20_sample, soil_blk_10_20_taken, bulk20_30_sample, soil_blk_20_30_taken, standard_sample, soil_std_taken, organic_sample, soil_org_depth_cando, soil_taken, soil_crop, photo_point, photo_north, photo_south, photo_east, photo_west, transect, revisit, th_gps_dist, file_path_gisco_north, file_path_gisco_south, file_path_gisco_east, file_path_gisco_west, file_path_gisco_point, gps_geom, th_geom, trans_geom, file_path_ftp_cover |

## 5 Automated identification of photos with potential privacy content

### 5.1 Check for the presence of privacy content and manual anonymisation

According to the guidelines of the LUCAS project, it must be ensured that no private content is included in the published images. This applies in particular to vehicle registration plates and recognisable persons and faces, which have to be blurred or removed in the photos. Since this anonymisation requirement applies equally to the previously unpublished cover photos, all cover photos must be checked for the presence of private content.

In order to fulfil this essential quality requirement for the image data, the checking of the images was carried out purely manually in previous LUCAS campaigns via a visual inspection of the photos. To reduce the manual effort and to improve the anonymisation quality, an automated procedure was used for the first time to support the image anonymisation process in the

65 LUCAS campaign 2018.

The method developed during the 2018 survey is based on the highly efficient Convolution Neural Network (CNN) YOLO (Redmon and Farhadi, 2018). This neural network enables the recognition of a large number of different object classes as well as multiple objects per image at a very high speed.

In parallel to the established manual control procedure, the CNN approach was independently tested. It confirmed that YOLO

ideally fulfils the basic requirements for the task of pre-classification and can be used as a binary classifier with the classes (1:anonymisation potentially necessary / 2:anonymisation not necessary).

Essential for such an approach is the guarantee of a low false negative rate of the automated binary pre-classification by the CNN, i.e. the acceptance of a low specificity during the first step of this two-step procedure. This was achieved by a suitable choice of object classes from the pool of all available classes and the use of a suitable detection threshold (0.1).The subsequent

manual control in the second step ensures an almost vanishing false positive rate and thus a high sensitivity (true positive rate, recall or hit rate).

Thus, this two-step approach maximises specificity and sensitivity and achieves a very high overall accuracy of image anonymisation. The cover photos analysed in this study do not differ technically from the previously anonymised LUCAS images, and thus the described procedure could be applied directly to the cover photos without modification.

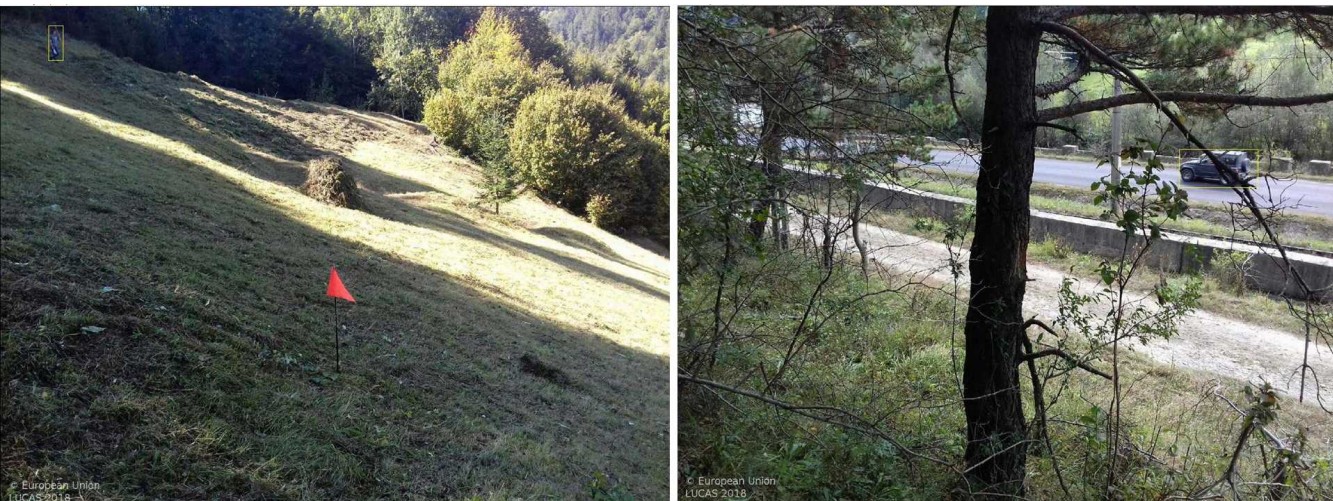

**Figure 3.** Examples showing correct identification of potential anonymisation elements (human person left and car right), which however needed no anonymisation due to lacking face or number plate recognition





## 5.2 Results of the anonymisation check

This two-step anonymisation approach from the 2018 campaign was also applied to the LUCAS cover photos from the campaigns 2006-2018. Table 2 provides an overview of the photo anonymisation checks for all LUCAS campaigns. In total, 875,661 were processed, out of which 66 were corrupted, meaning they failed to ingest. In turn 16,880 photos or 1.92% were flagged by the CNN as containing elements with potential anonymisation need. Finally, a total of 346 were manually shortlisted from the previously flagged set as actually containing elements worth anonymising for. Added to these are the 604, which did not have a match within the LUCAS harmonized product (d'Andrimont et al., 2020). The difference between columns five (i.e. 1st step) and six (i.e. 2nd step) shows that the number of photos containing potential non-anonymized elements was reduced by ca. 97.5 %-99% for each campaign, i.e. only roughly 1-2.5% of the photos have to be checked visually. Still, within this reduced amount, most photos will not contain elements that need to be anonymised. The main known reasons why photos are marked as containing potential non-anonymised elements, while not containing any, are listed below:

1. Photo is already anonymised: During the LUCAS campaigns strict anonymisation procedures were in place to cover persons and car plates with white bars. Despite this the neural network will detect a car as a car, even if its car plate is already correctly anonymised, i.e. covered with a white bar.

2. Photo shows only very small elements: Very small elements, i.e. cars or persons in the background of an image, which are not recognisable, do not have to be anonymised. Nevertheless, the neural network will detect these small cars, trucks, persons, etc. and thus mark the photos as containing potential non-anonymised elements.

3. Photo shows only a non-recognisable part of a car, person: The neural network detects objects, even if it is only partially visible. It therefore marks photos as containing potential non-anonymised objects, even if only a hand, an arm or a door of a car is visible, which do not have to be anonymised.

4. Wrongly classified objects: The neural network wrongly classified images as containing a certain element, which they did not, e.g. an animal or plant, classified as a person. The priority of the classification process was to not miss any (or as few as possible) non-anonymised objects. Therefore the object threshold score was set to a very low value of 0.1 to avoid missing any non-anonymised elements. Albeit ensuring this, the very low threshold score also delivers wrongly classified objects.

In total 1,016 images were removed from the original source set, which include the ones from columns *"no harmo"*, *"2nd step"*, and *"corrupted"*.




**Table 2.** Results of the photos screening with potential anonymisation issues. *source* specifies the total number of LUCAS cover images on disk; *no harmo* indicates LUCAS cover images on disk that do not have a corresponding row in the LUCAS harmonised product (d'Andrimont et al. (2020)); *no exif* is for images on disk that do not have any EXIF infomation encoded; *1st step* are images flagged by the YOLO network; *2nd step* are images flagged manually from the YOLO set as having some element, which is subject to anonymisation; *corrupted* are the images that have a visual distortion that makes the image unusable; the final column shows the images per year that are part of the published set.

| Year | Total number of photos surveyed | | | Flagged photos | | | Total number of photos published |
|---|---|---|---|---|---|---|---|
| | source | no harmo | no exif | 1st step | 2nd step | corrupted | |
| 2006 | 107,140 | 54 | 3 | 1,007 | 63 | 1 | 107,022 |
| 2009 | 150,125 | 39 | 9,559 | 2,239 | 80 | 13 | 149,993 |
| 2012 | 204,944 | 0 | 9,652 | 3,930 | 88 | 50 | 204,806 |
| 2015 | 217,638 | 0 | 1,654 | 4,654 | 28 | 1 | 217,609 |
| 2018 | 195,814 | 511 | 2,085 | 5,050 | 87 | 1 | 195,216 |
| **Total** | 875,661 | 604 | 22,953 | 16,880 | 346 | 66 | 874,646 |

## 6 Harmonisation of the filename and watermark

A number of changes had to be done to the filename and directory tree in order to harmonise both between the survey years. Namely, certain years had listed countries with a different country code (eg. "GR" instead of "EL"; or "GB", instead of "UK"), filenames had used an upper or a lower case for either the "c" to indicate the "cover" status of the LUCAS image, or differences in the ".jpg" extension. To coherently catalogue this, all images were renamed to fit the convention "LU-CASxxxx_PointID_Cover.jpg". Additionally, watermarks were added to the 2009 and 2012 images in cases where these were lacking.

## 7 Final Data Overview

The distribution of LUCAS Cover photos per land cover and per year is shown in Figure 5. The distribution per country is presented in Table 3. For each survey year, a number ranging from 107,022 (in 2006) to 217,609 (in 2015) were collected totaling 874,666 for the five surveys. The sampling of LUCAS aims to revisit some of the points in successive surveys resulting thus to point revisit for each point surveyed ranging from one to five as shown in Figure 4. In Figure 6, one random photo example is shown for each land cover class.



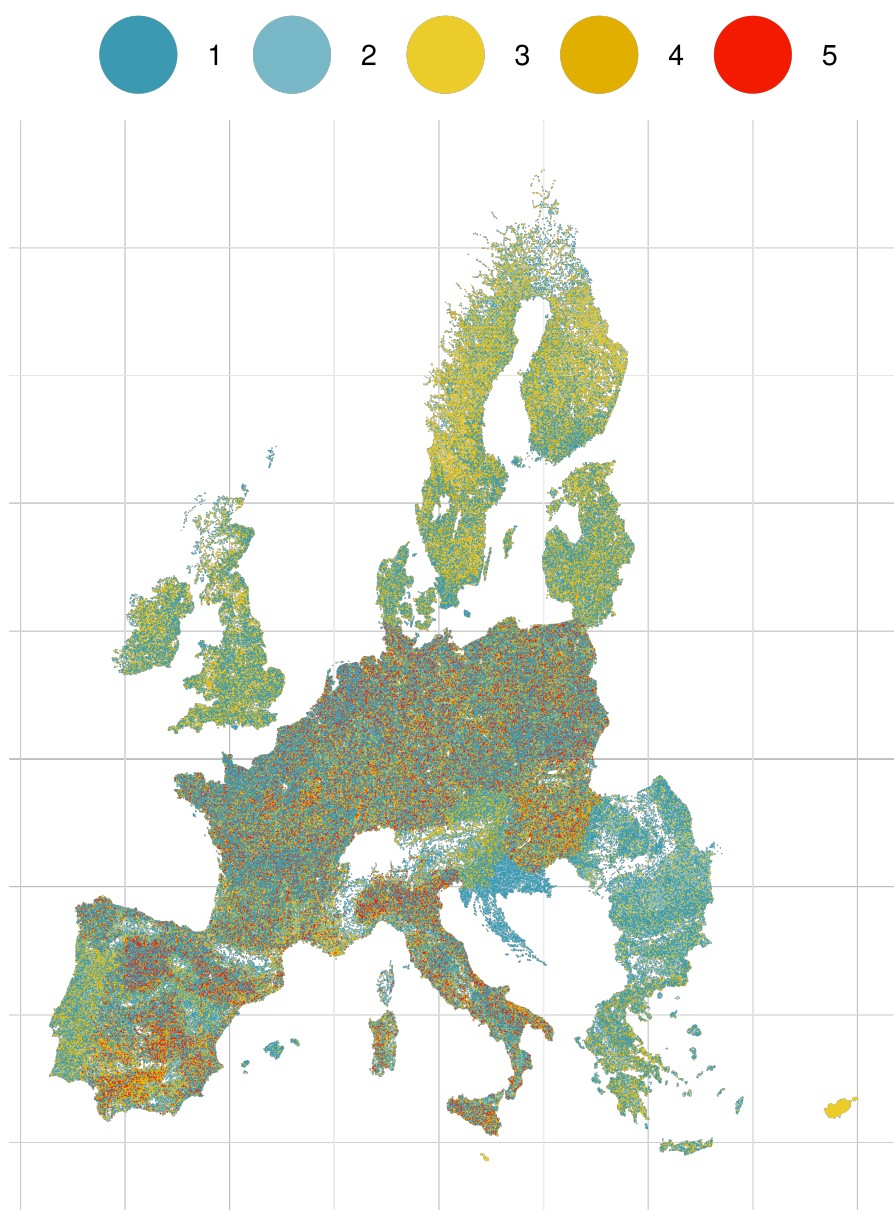

**Figure 4.** LUCAS Cover photos geographical distribution and number of (re)visits to each survey over the five surveys between 2006 and 2018. Visits range from one to five. Map rojection: EPSG 3035.





**Table 3.** Number of LUCAS Cover photos per country and per year.

|  | 2006 | 2009 | 2012 | 2015 | 2018 | Total # |
|---|---|---|---|---|---|---|
| **AT** | 0 | 3571 | 5249 | 4965 | 3659 | 17,444 |
| **BE** | 1898 | 1066 | 1856 | 1907 | 2187 | 8914 |
| **BG** | 0 | 0 | 5794 | 5226 | 4977 | 15,997 |
| **CY** | 0 | 0 | 1045 | 1127 | 1136 | 3308 |
| **CZ** | 4568 | 4230 | 5097 | 5068 | 4779 | 23,742 |
| **DE** | 5311 | 16,333 | 21,497 | 21,364 | 19,765 | 84,270 |
| **DK** | 0 | 1881 | 2857 | 3026 | 2610 | 10,374 |
| **EE** | 0 | 1507 | 1773 | 1806 | 1418 | 6504 |
| **EL** | 0 | 4461 | 5938 | 6202 | 6881 | 23,482 |
| **ES** | 23,451 | 18,762 | 28,447 | 28,623 | 25,525 | 124,808 |
| **FI** | 0 | 10,230 | 9281 | 10,422 | 6545 | 36,478 |
| **FR** | 29,187 | 22,614 | 31,005 | 32,674 | 31,570 | 147,050 |
| **HR** | 0 | 0 | 0 | 2674 | 2053 | 4727 |
| **HU** | 6657 | 4377 | 4124 | 4047 | 3276 | 22,481 |
| **IE** | 0 | 1414 | 2336 | 2495 | 2505 | 8750 |
| **IT** | 12,315 | 10,345 | 15,086 | 15,230 | 15,131 | 68,107 |
| **LT** | 0 | 2780 | 3491 | 3420 | 2608 | 12,299 |
| **LU** | 156 | 133 | 178 | 181 | 210 | 858 |
| **LV** | 0 | 2543 | 3341 | 3669 | 2826 | 12,379 |
| **MT** | 0 | 0 | 43 | 50 | 48 | 141 |
| **NL** | 2479 | 1652 | 1609 | 1693 | 3122 | 10,555 |
| **PL** | 18,433 | 14,709 | 19,059 | 18,963 | 18,064 | 89,228 |
| **PT** | 0 | 3848 | 6018 | 5714 | 4624 | 20,204 |
| **RO** | 0 | 0 | 641 | 10,385 | 8473 | 19,499 |
| **SE** | 0 | 13,624 | 16,876 | 14,646 | 9556 | 54,702 |
| **SI** | 0 | 1036 | 1381 | 1440 | 1382 | 5239 |
| **SK** | 2567 | 2230 | 2066 | 2101 | 1760 | 10,724 |
| **UK** | 0 | 6642 | 8709 | 8475 | 8526 | 32,352 |
| **NOT EU** | 0 | 5 | 9 | 16 | 0 | 30 |
| **Total # records** | 107,022 | 149,993 | 204,806 | 217,609 | 195,216 | 874,646 |



**Figure 5.** Distribution of LUCAS Cover photos in land cover classes in the multi-year harmonised LUCAS database. In cases where survey years are not present please orientate oneself with reference to adjacent classes of the same color. Counting for the distribution of each class begins at 2018 and ends with 2006 due to the relative abundance of 2018 in terms of classes compared to other years.

**Figure 6.** Examples of LUCAS Cover photos for all classes (having at least 300 images per class). The land cover class is shown in the top left corner of the image in white. The letter at the top of the row represents the LUCAS level 1 label (variable *letter_group*). See Figure 5 for label correspondences.



## 8    Limitations

Several limitations inherent to the dataset are briefly discussed. The first limitation is linked to the survey protocol, which is
not detailing the field of view required to take the photos. Indeed, the instruction to the surveyor is to make sure the plant
could be recognised on the picture which is subjective to the surveyor. As illustrated in Figure 6, the photos could be taken
with a diversity of views: whole plant views, landscape views (sometimes with sky), plants with artificial background (red
support), plants with bare soil. Also sometimes, the surveyor takes the plant from the soil and the pictures thus contain the root
system and vegetative organs. Future surveys could consider specifically collecting information on the view type. Applications
dedicated to recognizing plant species such as Pl@ntNet ask the surveyor to select the type of view as leave, flower, fruit, bark,
whole plant or other (Goëau et al., 2013).

    Another limitation is the lack of EXIF information for some photos as highlighted in Table 2. Also, the quality of the EXIF
data collected, when available, depends on the quality of the sensors and its calibration. Standardizing the type of sensors used
to collect the images would greatly facilitate this routine and the uptake of the data. Another option would be to hardcode some
variables when possible, such as looking direction from time of day and year and the angle of shadows when sunny.

## 9    Potential use of the data and perspectives

The LUCAS Cover dataset with systematically sampled geo-located observations and photos of crops, trees, shrubs, grasses,
and other plants, can be the source for different uses and drive the development of various applications. The specific advantages
are 1) the sample design where the regular systematic 2-km LUCAS grid ensures an exhaustive EU-wide coverage, 2) the
observations were done during a period of 15 years with several of the points revisited up to 5 times providing a unique
historical perspective, 3) the photos are annotated with a label following the LUCAS legend, and 4) while this label may not be
precise enough for various applications, computer vision based methods could extract information from the image and enrich
the label. . Indeed, LUCAS has been designed to collect statistics about land use and land cover, and specific applications will
have different needs. The precision of the legend is for example not sufficient for botanical applications that need species level
information on observed plants. However, the dataset could thus provide training data to build deep learning convolutional
neural networks to recognize and classify trees, plants, and crop types along with their phenological stages on photos, such as
in d'Andrimont et al. (2022b).

Indeed, recent advances in combining photos sources from citizens and experts in combination with computer vision are si-
multaneously enabling species identification and the gathering of occurrence data (e.g. PlantNet (Goëau et al., 2018), iNaturalist
(Nugent, 2018), or Flora Incognita (Mäder et al., 2021)). Collection of such geo-located plant species occurrences contribute
to collaborative data platforms such as the Global Biodiversity Information Facility (GBIF). This can be complimentary to
long-term and high-quality but resource-intensive biodiversity assessments by professional botanists (Miller-Rushing et al.,
2012). The photos in this new LUCAS cover photo dataset could be ingested in such applications. In fact, the LUCAS Cover
photos of crops (letter group "B") are currently used to generate a specific application within Pl@ntNet to recognize crops.



## 10 Conclusions

The LUCAS surveys have resulted in the most comprehensive in-situ database on land cover and land use in the EU. While close-up photos of the land cover had been collected for most of the in-situ points, they had never been published. This data paper represents an effort to organize, document, curate, and publish this dataset following FAIR principles. This resulted in 874,646 geo-located photos along with surveyed information on land cover and land use following LUCAS legend level 2, inherited from the attribute information of the LUCAS harmonised product. The LUCAS Cover photos and dataset can feed various applications and developments relying on recent advances in geo-spatial analysis and statistical learning.

## 11 Data availability

This section describes the data-set provided along with this manuscript including the tables and photos. The data is available at https://jeodpp.jrc.ec.europa.eu/ftp/jrc-opendata/LUCAS/LUCAS_COVER/ , d'Andrimont et al. (2022a).

1. **Photos** The 874,646 LUCAS Cover photos are available on the FTP : Downloadable here https://jeodpp.jrc.ec.europa. eu/ftp/jrc-opendata/LUCAS/LUCAS_COVER/

2. **Tables** (exif and LUCAS harmonised) https://jeodpp.jrc.ec.europa.eu/ftp/jrc-opendata/LUCAS/LUCAS_COVER/tables

   **LUCAS cover exif table** ( *lucas_cover_exif.csv* ). The table contains 82 variables described in Table 1.

   **LUCAS harmonised cover table** (*lucas_harmo_cover_attr.csv*). The table contains 121 variables described in Table 1.

*Author contributions.* All the authors processed and analyzed the data, wrote the paper, provided comments and suggestions on the manuscript. ESTAT designed the survey methodology. EFTAS and Imaging Consulting carried out the anonymization process of the LUCAS Cover photos.

*Competing interests.* The authors declare that they have no known competing financial interests or personal relationships that could have appeared to influence the work reported in this paper.

*Acknowledgements.* The authors would like to express their appreciation to all that have been involved in the LUCAS surveys and especially to the LUCAS surveyors for their careful observations that underpin the value of the resulting LUCAS datasets.



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
