# Peer review of "LUCAS Cover photos 2006-2018 over the EU: 874,646 spatially distributed geo-tagged close-up photos with land cover and plant species label"

_Earth System Science Data, 2022_

## Author Response (AR1)

*Title: LUCAS Cover photos 2006-2018 over the EU: 874,646  spatially distributed geo-tagged close-up photos with land cover and plant species label*

*Author(s): Raphaël d'Andrimont et al.*

*MS No.: essd-2022-117*

*MS type: Data description paper*

**REVIEWER 1:**

General Comments:

This paper presents an interesting dataset of geo located photos of land Cover in the EU that was unpublished until now. This dataset is complementary to an existing database of photos from the LUCAS survey. This new dataset (are related paper) is worth to be published.

The abstract and the description of the differences between the photos from the already published dataset and the photos from this new dataset can be improved (see next section)

*Thank you for the positive review and the comments that we have addressed below.*

**Technical corrections**

The abstract would need to be improved. In particular the abstract cannot be easily understood without reading the manuscript itself. E.g.

- The First sentence in abstract seem to lack a word. Suggestion to add 'exercise' as follows: "In the European Union, a tri-annual surveyed sample **exercise** collects land cover .."

*We've re-written the sentence, which now reads:*

*"In the European Union, a tri-annual survey samples land cover and land use information under the Land Use/Cover Area frame Survey since 2006."*

- The difference between "landscape and point photos" is not obvious (i.e. one has to read the manuscript to get the information)

*We've re-written the sentence, which now reads: "A total of 1,351,293 observations at 651,780 unique locations for 106 variables were collected during the five LUCAS surveys, including a total of 5.4 million landscape photos, representing the observer view in the four cardinal directions, and point photos showing the actual surveyed point."*

- Difference between "landscape and point photos" and "LUCAS Cover photos" is not obvious from the abstract alone.

*While it is hard to discuss in depth the difference between the types of photos, collected during the LUCAS surveys in the abstract, we have, nevertheless tried to make it more understandable by re-writing the sentences as follows:*

*"In addition to these previously published photos, additional LUCAS cover photos were recorded, showing a close-up view and thus more detail of the sampled tree, crop and plant species."*

*Then the following sentence was deleted (i.e.""These photos contain more details so that tree, crop, and plant species should be identifiable")*

- "LUCAS Cover photos that show the relevant land cover in its entirety" What does it mean? What is the difference or specificity compared to the landscape photos?

*Sentence re-written to:*

*"Between 2006 and 2018, 875,661 LUCAS Cover photos were collected that show the relevant land cover on the entire photo with the absence of any other elements of the landscape in the frame."*

*The difference, in essence, is that the landscape photos, as the name suggests, show the landscape view in the cardinal direction. What this means is that there are other elements on the image, other than the land cover – trees, sky, etc. The Cover photos show only the land cover so that it can be used directly, without cropping or detections, in, let's say – image classification workflows.*

- "Due to surveyor differences, the images sometimes display elements that require a two-stage deep learning anonymisation process". The rationale of the 'surveyor differences' is not obvious to understand the need of a ' anonymization process'

*We've changed the sentence to include also:*

*"Photos containing potential privacy content were identified following a two-stage deep learning anonymisation process (346 photos were removed before publication)."*

*In reality these are the reasons why anonymization is needed on the Cover photos. Either the surveyor is not well trained on what they need to image, or the situation on the field doesn't permit a different camera angle.*

- From section 3 and figure 2 the Difference between landscape photos and "LUCAS Cover photos is quite clear. But the difference between landscape and point photos is still not evident.

*Yes, indeed, you are very correct that the landscape and point photos look sometimes very similar. Yet, the difference is supposed to be in the depth of the photo – landscape photos are supposed to look at the landscape in depth to the supposed horizon line, while the point photo is supposed to image the surveyed point. The difference is subtle at first glance on a single*

*example, yet substantial when you start work with the data. For example, colleagues doing so are using specifically the LUCAS point photo to perform semantic segmentation in order to detect the land covers present on the photo at the point location (Martinez et al, coming up). In this sense using the landscape photos in the cardinal direction proved unsatisfactory, as they show the land cover in the respective depth in the respective direction. Strictly speaking, this shouldn't be explained in this paper, but in the one documenting the LUCAS harmonization, as it already is. We've included a sentence in the paragraph linking the information:*

*"This P, N, E, S, W photo dataset corresponds to 5440459 photos for the five surveys. These photos are publicly available for download along with an EXIF database in d'Andrimont et al. (2020), containing image metadata, and an explanation as to the difference between landscape and point photos."*

*The previous sentence:*

*"Cardinal direction photos are taken for each observed in-situ point into Point, North, East, South, and West (P, N, E, S, W) direction (example of these are the five first photos from left to right in Figure 2)."*

*was changed to*

*"Photos are taken for each observed in-situ point covering the actual Point (P) and the four cardinal direction views North, East, South, and West (P, N, E, S, W) (example of these are the five first photos from left to right in Figure 2)."*

**REVIEWER 2**

Geo-located photos are increasingly used in the land cover mapping and biodiversity studies. This study presents a large collection containing 874k geo-referenced photos taken at during the period between 2006-2018. Being different from the previously published LUCAS survery photos, the Cover photo is taken at very close distance to the vegetation so that the leaves, structures of the plants can be clearly recorded. The dataset is undergone through an anonymization filtering and some simple statistics of the data are shown. The dataset is clearly described and the potential applications is discussed. The manuscript can be published after some issues to be addressed.

*Thank you for the positive review and the comments that we have addressed below.*

1. The structure of how the photos is organized should be further explained. Currently using the link provided by the authors, under LUCAS_COVER, photos are organized by the years obtained. The subdirectory of the year is country, in the country folder, they are in folders named with three digits number, and in these folders, there are again folders named with three digits number. The file system structure and the meaning of these three digits numbers should be further explained.

*We've included a brief explanation of the file structure:*

*The directory tree for the photo database follows the standard of the original EUROSTAT data. It is thereby organised, starting from the folder shown in the provided link, descending*

*into a folder specifying the year of the survey ("LUCASYYYY"), followed by folders of each respective NUTS0 code that the point is located in, in turn proceeded by folders of the first three digits, and the second three digits of the point id. The JPG files that constitute the LUCAS Cover photos are to be found in the folder named after the second three digits of the point id, and named according to the convention described in Section 6.*

2. The quality of the photos varies greatly from site to site. The light condition, distance to the target various greatly. For example, LUCAS2012_49204454_Cover.jpg show very dark color and is taken with an upward looking direction. Limited information can be extracted from this photo. Some of the photos are not taken in the field but with samples collected and photos are taken either in a car or a lab (photos in LUCAS2012/FI/492/044). The geo-location of these photos should be checked to make sure that they are correctly labeled.

*Indeed, the image quality varies sometimes drastically between points, years, and surveyors. It was decided that we, as carrying out the harmonization, are not responsible for determining what constitutes a "good quality" image, as this depends to a large extent on the use case. What is a good image for one use case is not necessarily the same for another and vice-versa. We wanted to keep as much of the original data intact as possible and serve it to users, who would determine what works for their use case. The statement that images from LUCAS2012/FI/492/044 are taken from a car or in a lab seems incorrect. If the reviewer is referring to the black background images – this is the folder that the surveyor has with them when on the field, and are sometimes taking image against the background, which, for example, for a use case we are using the images for proved to be a bad quality image. The "images taken from a car" also seems implausible – they would be significantly more blurred if taken from a car. The geo-location of the images has been cross-checked where possible to extract the location from the EXIF.*

**Detailed comments:**

- L16, what is the difference between "points" and "locations"?

*Revised the sentence as:*

*During the five campaigns already carried out, a total of 1351293 point surveys at 651780 unique locations were performed along with 5.4 million landscape photos.*

- Figure3. are these cover photos? They are taken close enough to recognize the plant structure.

*These are not Cover images, the first panel on the left is a P (point) image, evidenced by the flag, showing the exact point location or nearest location reached, and the one on the right is an image in a cardinal direction. The figure is provided from the EFTAS sub-contractor, hired by ESTAT to carry out the anonymization, as done on the main LUCAS photo archive, which is the same workflow, carried out on the LUCAS Cover photos.*

L86, what are the columns? Are they a from a table?

*Sentence corrected as:*

*"The difference between columns five (i.e. 1st step) and six (i.e. 2nd step) of Table 2  shows ..."*

L112, maybe explain what does xxxx represent in the file name?

*Sentence corrected as:*

*"To coherently catalogue this, all images were renamed to fit the convention "LUCASYYYY\_PointID\_Cover.jpg", where YYYY is a placeholder for the year of the survey."*

Figure4, it would be helpful to add the national boundaries as the background.

*NUTS0 boundaries added to plot.*

Figure5. is it possible to provide more specific classifications (species) of the woodland and shrubland?

*Figure 5 shows the legend classes in LUCAS legend level 3 (LC1 variable), it is possible to descend one level further through the LUCAS legend hierarchy - level 4 (LC1_SPEC variable), where different classes of woodland and shrubland are listed, but then the figure becomes illegible. Simply put, all level-4 classes from the other letter groups put together become in the hundreds. The figure as it stands is the most worthwhile representation of the data in a figure of this kind.*

Figure6. I do not see photos in categories such as G10, G21, F10, F20, is it because there are less than 300 photos for this class? I would suggest the authors show at least 1 for all classes.

*Updated the figure to include all classes (not just the >300 images per class).*